# Structures of the wild-type MexAB–OprM tripartite pump reveal its complex formation and drug efflux mechanism

Kenta Tsutsumi [1], Ryo Yonehara[1], Etsuko Ishizaka-Ikeda[1], Naoyuki Miyazaki [1], Shintaro Maeda[1,2], Kenji Iwasaki[1,3], Atsushi Nakagawa [1] & Eiki Yamashita [1]

In *Pseudomonas aeruginosa*, MexAB–OprM plays a central role in multidrug resistance by ejecting various drug compounds, which is one of the causes of serious nosocomial infections. Although the structures of the components of MexAB–OprM have been solved individually by X-ray crystallography, no structural information for fully assembled pumps from *P. aeruginosa* were previously available. In this study, we present the structure of wild-type MexAB–OprM in the presence or absence of drugs at near-atomic resolution. The structure reveals that OprM does not interact with MexB directly, and that it opens its periplasmic gate by forming a complex. Furthermore, we confirm the residues essential for complex formation and observed a movement of the drug entrance gate. Based on these results, we propose mechanisms for complex formation and drug efflux.

[1] Institute for Protein Research, Osaka University, Suita 565-0871 Osaka, Japan. [2] Present address: The Scripps Research Institute Department of Integrative Structural and Computational Biology, North Torrey Pines Road, La Jolla, CA 10550, USA. [3] Present address: University of Tsukuba Life Science Center for Survival Dynamics, Tsukuba Advanced Research Alliance 1-1-1 Tennodai, Tsukuba 305-8577 Ibaraki, Japan. These authors contributed equally: Kenta Tsutsumi, Ryo Yonehara. Correspondence and requests for materials should be addressed to E.Y. (email: eiki@protein.osaka-u.ac.jp)

Gram-negative bacteria have a strong outer membrane consisting of lipid-bilayer and peptidoglycan in addition to an inner membrane, and they express several types of tripartite efflux pumps that penetrate both membranes and release foreign substances[1]. Multidrug efflux pumps of the resistance–nodulation–cell division (RND) superfamily, which are expressed specifically in Gram-negative bacteria, consist of an RND transporter that penetrates the inner membrane and plays a major role in drug efflux via a proton gradient; an outer membrane factor (OMF) that penetrates the outer membrane and secures the efflux route for drugs; and a membrane fusion protein (MFP) that is anchored to the inner membrane and connects the RND transporter and OMF[2,3]. Overexpression of RND-type multidrug efflux pumps is a primary cause of multidrug resistance[4]. *Pseudomonas aeruginosa*, a common Gram-negative bacterium, causes nosocomial infections and has particularly high drug resistance[5]. MexAB–OprM, the only constitutively expressed pump in *P. aeruginosa*, is thought to contribute significantly to drug resistance in this species[6]. The crystal structures of the three proteins constituting MexAB–OprM, MexA (MFP), MexB (RND), and OprM (OMF), have already been solved[7–12].

OprM forms a trimer with threefold symmetry, and two coiled-coils of helices 3 and 4 (H3–H4) and helices 7 and 8 (H7–H8) in the α-barrel domain form a gate on the periplasmic side. Because H7–H8 blocks the gate in the crystal structure, it is thought that the gate opens by interacting with the MFP[11]. MexA consists of four domains: the membrane-proximal (MP), β-barrel, lipoyl, and α-hairpin domains[8]; it is thought that MexA forms a hexamer in MexAB–OprM. MexB is composed of three domains: a transmembrane (TM) domain with 12 α-helices, a porter domain containing the drug-binding pocket, and a funnel-like (FL) domain involved in interactions with other components[9]. A porter domain further consists of four sub-domains; PN1, PN2, PC1, and PC2, which translate and rotate during the transport process, thereby altering drug accessibility. MexB forms an asymmetric trimer in crystal structure, and it is thought that each protomer shifts conformational states (Access, Binding, and Extrusion) during drug efflux, similar to the functional rotation mechanism proposed based on the structure of the *Escherichia coli* homolog AcrB[13]. On the other hand, based on structural analysis of an RND pump derived from *Campylobacter jejuni*[14], an distinct efflux mechanism was proposed, suggesting that the mechanisms of action of RND family members may have diverged.

AcrAB–TolC, a major RND-type multidrug efflux pump in *E. coli*, has been extensively investigated, and its structure was solved by cryo-electron microscopy (cryo-EM)[15,16]. These studies revealed that OMF and the RND transporter do not directly interact. Furthermore, recent work revealed the asymmetric structure of AcrAB–TolC with closed or open TolC at medium resolution, as well as the symmetric structure of open AcrAB–TolC in the presence of inhibitor, at near-atomic resolution[17]. Based on these structures, it was suggested that AcrAB–TolC initially forms a complex in the closed state, and then TolC opens via rearrangement of the AcrA hexamer induced by a conformational change in AcrB. These studies used genetically engineered or disulfide linked MFP-RND fusion proteins for structural analysis. Although 3D structure of MexAB–OprM using a negative-stain method was reported previously[18], no (near-) atomic resolution structure of a multidrug efflux pump derived from *P. aeruginosa* has been elucidated to date. Here, we report a structure of the wild-type multidrug efflux pump MexAB–OprM from *P. aeruginosa* at near-atomic resolution, in the presence or absence of drug. Based on a structural comparison, we propose a mechanism for complex formation, as well

as a mechanism for drug release that differs from previously proposed mechanisms for RND-type multidrug efflux pump complexes.

## Results

**Structural determination and overall structure of MexAB–OprM.** Fully assembled MexAB–OprM pump was prepared by in vitro reconstruction method (Supplementary Fig. 2). The detergents used in each preparation of MexB and OprM were replaced with amphipol A8–35[19]. The three-dimensional (3D) structure of MexAB–OprM was determined by cryo-EM single-particle analysis (Supplementary Fig. 3a). During this analysis, we identified two modes of binding of OprM to MexA; accordingly, we determined the structures of both states of MexAB–OprM pumps at resolutions of 3.64 Å (state A) and 3.76 Å (state B). These maps are of sufficient quality to $C_α$-trace, and we could identify orientations of almost all of bulky side chains (Supplementary Figs. 3, 10).

The overall structure of MexAB–OprM is a vertically elongated rod shape, ~320 Å along the long axis and ~110 Å along the short axis, and the stoichiometry of OprM, MexA, and MexB is 1:2:1 (Fig. 1). MexB does not directly contact OprM; instead, MexA joins MexB and OprM by forming a funnel-like hexamer. Among the four domains of MexA, only the α-hairpin domain interacts with OprM; the β-barrel domain interacts with the FL domain of MexB, and the MP domain interacts with the porter domain of MexB. The lipoyl domain of MexA does not connect with either OprM or MexB, but it forms a hexameric ring along with the β-barrel domains. MexA protomers are divided into two classes according to the area of their interaction surface with MexB: the large-contact (LC) protomer (~1450 Å²) and the small contact (SC) protomer (~950 Å²). The MP domains of the LC protomers are located in the PC1 domains of each MexB protomer (Fig. 1, colored by orange), whereas the MP domains of the SC protomers were located between each protomer of MexB (Fig. 1, colored in magenta). Each protomer of MexB is in one of three different states in the crystal structure (Supplementary Fig. 7). On the other hand, the periplasmic gate of OprM is opened and the drug efflux route is secured unlike the crystal structure. State A and state B are quite similar (Cα RMSD: 0.47); however, the relative positions of OprM in each state are related by a 60° rotation. Hereafter, we denote state A as the 0° state and state B as the 60° state.

**Interaction between OprM and MexA.** Local resolution validation shows that the contact face region between OprM and MexA has ~3 Å resolution, which allows us to fit good atomic models into the maps (Supplementary Fig. 3e, h). In the 0° state, H3–H4 of OprM face the α-hairpin tips of the LC protomers, whereas H7–H8 face the α-hairpin tips of the SC protomers. In the 60° state, on the other hand, H3–H4 of OprM face the α-hairpin tips of the SC protomers, and H7–H8 face the α-hairpin tips of the LC protomers (Figs. 1 and 2). The Cα RMSD between periplasmic tips of the OprM protomer in two states (S188–E214: H3–H4; Y396–F422: H7–H8) is 1.1 Å, and the Cα RMSD between α-hairpin domains of hexameric MexA in two states (A74–F134) is 0.33 Å. The contact surface areas between the MexA hexamer and OprM trimer are 2992 Å² (0° state) and 2983 Å² (60° state). Several hydrogen bonds are present between main chains: Q104 in MexA binds to A203 or Y411 in OprM, and K108 in MexA binds to G199 or G407 in OprM (Fig. 2a, b; Supplementary Fig. 11). Also, L100 in MexA engages in hydrophobic interactions with V198 and V200 on H3–H4 or V408 on H7–H8. Moreover, the side chain of R403 in OprM extends inward from H7 and seems to form a hydrogen

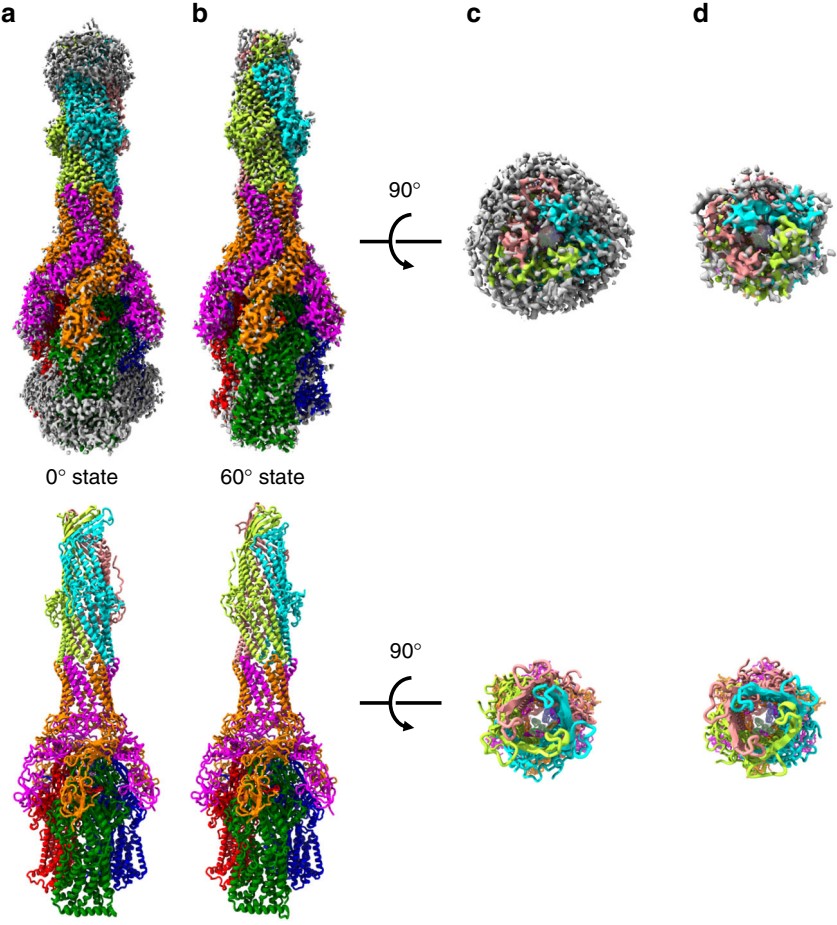

**Fig. 1** Overall structure of MexAB–OprM. Single-particle cryo-EM reconstruction (top) and model of MexAB–OprM (bottom) in the 0° state (**a**) or 60° state (**b**) viewed from the periplasmic space (**a**, **b**) or from the outside the cell (**c**, **d**). The OprM protomers are colored in cyan, salmon, and lemon. MexA protomers are colored in magenta (SC protomer) or orange (LC protomer). MexB protomers in the Access, Resting, and Extrusion states are colored in green, blue, and red, respectively

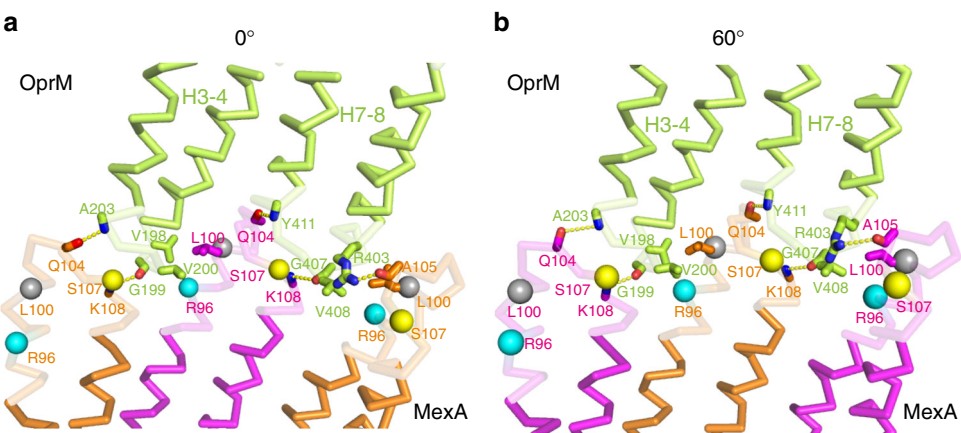

**Fig. 2** Interaction between OprM and MexA. Close-up views of the α-hairpin of OprM (shown in lemon) and the α-hairpin of MexA [shown in magenta (SC protomer) or orange (LC protomer)] in the 0° state (**a**) or 60° state (**b**). Previously proposed RLS motifs are shown as balls colored in cyan (R96), gray (L100), and yellow (S107)

bond with the main chain of A105 in MexA. These results suggest that the two modes of binding between MexA and OprM do not differ significantly. In addition, the number of particles in the 3D classification of both states was not much different (Supplementary Fig. 3), and it might be considered that these two binding modes exist equally within bacterial cells.

Regarding the interaction between OprM and MexA, several residues that were thought to be important for interactions, as well as a few models of complex formation, have been proposed based on previous mutation experiments and molecular dynamics (MD) simulations of tripartite efflux pumps[20–25]. To identify the residues involved in complex formation, we performed a functional analysis

for residues previously proposed to be required[21,24,25] or that seemed important based on our cryo-EM structure. Specifically, we performed in vitro complex formation analysis using site-point mutation and size exclusion chromatography (SEC) (Supplementary Fig. 5) and conducted experiments to determine the in vivo drug resistance (Supplementary Fig. 6). Alanine mutations for G199 and G407 in OprM, which were proposed to be critical residues in a previous study[24], completely abolished complex formation. These glycine residues are located at equivalent positions between the H3–H4 and H7–H8 loops, and the main chains of these glycine residues and the K108 residue of MexA were at a distance suitable for formation of a hydrogen bond (Fig. 2). Aspartate mutation of the L100 residue of MexA, one of the RLS motifs proposed to be important for complex formation of MFP based on structural analysis of AcrAB–TolC[16,21], also completely abolished complex formation and drug resistance (Supplementary Fig. 5c, 6). By contrast, aspartate mutation of the adjacent residue L99 had no effect on complex formation, highlighting the importance of L100. Because L100 contacts H3 or H7 of OprM from the side, and its side chain is close to V200 on H3 or V408 on H7 (Fig. 2), we speculated that L100 binds MexA and OprM via hydrophobic interactions. R403 of OprM, not previously proposed to be important, was critical for complex formation: alanine mutation at this position abolished complex formation (Supplementary Fig. 5d). S107 in MexA, one of the RLS motifs, was also proposed to be an important residue based on MD simulation[25]. Although aspartate mutation of this residue disrupted complex formation and decreased drug resistance in the drug-resistance assay, we could confirm no specific interaction of S107 in our structure. The side chain of S107 is located at the narrow gap between MexA tip and OprM tip (4–5.5 Å), so the inhibition of complex-formation ability by the aspartate mutation is thought to be due to steric hindrance of the side chain. Like L100 and S107, the remaining RLS motif, R96, is also conserved, but its alanine mutants formed a complex as efficiently as the wild-type. By contrast, aspartate mutation at this residue decreases its minimum inhibitory concentration experiments[21], consistent with our experimental results of the drug-resistance assay (Supplementary Fig. 6), and also diminished complex formation. Based on these results, we can conclude that R96 in MexA is not essential for complex formation, and that the decrease in complex-forming ability caused by the aspartate mutation was due to charge repulsion. Furthermore, D103 and Q104 in MexA were predicted to be important for complex formation based on previous MD simulations[25]; however, alanine mutations of these residues did not show significant effects in either in vitro or in vivo functional analysis (Supplementary Fig. 5c, 6). Therefore, the tip-to-tip interaction of MexA and OprM is based mainly on the interaction between the main chains, such as Q104–A203 or –Y411 and the aforementioned K108–G199 or –G407 (Fig. 2). In addition, the interactions between the side faces of OprM and MexA, such as the hydrophobic interaction centering on L100 in MexA or the hydrogen bond between the side chain of R403 in OprM and the main chain of A105 in MexA, are also essential for complex formation.

**Interaction between each MexA protomer.** Except for the MP domains, the MexA hexamer has C6 symmetry, and is formed by arrangement of the β-barrel domains in a ring shape via electrostatic interactions (area of contact surface: 1254 Å²). For example, R39 or R147 interacts with E152 or E226 of the adjacent protomer (Fig. 3a). Repelling charge mutations (R39D or R147D) prevented complex formation, as determined by SEC experiments (Supplementary Fig. 5b), and decreased drug resistances in the drug-resistance assay (Supplementary Fig. 6). The hexameric assembly in the lipoyl and β-barrel domains were similar to those

in MacA, an MFP of the ABC-type multidrug efflux pump MacAB–TolC in *E. coli*, which forms hexamer in the single-crystal structure[26] and in the tripartite complex[27].

**Interaction between MexA and MexB.** MexA and MexB have two interacting faces: the β-barrel domain of MexA with the FL domain of MexB and the MP domain of MexA with the porter domain of MexB. The interaction area between the β-barrel and FL domains is about 600 Å², and did not differ significantly between the SC and LC protomers. However, the area of the MP domain and porter domain was significantly different: ~360 Å² for the SC protomer vs. ~860 Å² for the LC protomer. Note that there is little difference in these interactions regardless of the states of MexB.

For complex formation between the SC protomer and MexB, we observed two specific interactions in our cryo-EM structure. The key loop (L252–V260), located at the top of the FL domain of MexB, shifts onto the side of the adjacent protomer in comparison with the crystal structure, and sticks in the hollow formed by the β-barrel domain of the SC protomer (Fig. 3b). The side chains of R34 and T233 in this hollow are positioned at a distance suitable for formation of a hydrogen bond, with the carbonyl groups of P255 or N254 and S258 located in the key loop. In addition, the side chain of R277 in the MP domain is about 3–4 Å from the side chain of E244 in MexB, sufficient to form a hydrogen bond (Fig. 3c). Indeed, the alanine or aspartate mutant of R34 or R277, as well as the alanine or valine mutant of T233, lost the ability to form a complex (Supplementary Fig. 5a, e). These mutations also decreased drug resistance, as determined by the drug-resistance assay (Supplementary Fig. 6). Therefore, we can conclude that these three residues are essential for SC protomer binding. On the other hand, in the LC protomer, the entire MP domain faces PC1 of MexB, forming an interaction across the entire plane (Fig. 3d). PC1 undergoes no conformational change relative to the crystal structure except for a shifted helix (M653–A661), which was pushed out by F328 in the LC protomer (Fig. 3e).

**Comparison of the closed and open structures of OprM.** In the crystal structure of isolated OprM (PDB ID: 3d5k; https://doi.org/10.2210/pdb3D5K/pdb), the periplasmic gate is closed by the hydrophobic interaction of L412 between the protomers and the salt bridge between D416 and the adjacent R419[12]. In the protomers, the side chains of S188 and T192 on H3 form a hydrogen bond with R405 on H7. In our cryo-EM structure, H3–H4 opens ~9° compared with the crystal structure with T178 and Q222 acting as the fulcrum, and the hydrogen bonding with the side chain of R405 disappeared (Fig. 4a, b; Supplementary Movies 1, 2). Also, H7–H8 rotated ~19°, with R376 and R432 as the fulcrum. Consequently, the Cα atom of L412 moved 11.4 Å outward, and the salt bridge between R419 and D416 disappeared.

**The crystal structure and cryo-EM structure of MexB.** In our structures, MexB is asymmetric: specifically, the structures of the FL domain (excluding the key loop) and the TM domain were very similar to the Access, Bind, and Extrusion states in the previously reported crystal structures (Supplementary Fig. 7). By contrast, with respect to the porter domain, although the Access and Extrusion states are very similar, except for shifted helixes, a slight difference can be observed in the protomer corresponding to the Bind state. Compared with the crystal structure, PC2 is shifted by ~3 Å toward PC1 and the cleft between PC1 and PC2 does not exist (Fig. 5a, b). Also, because PC2 approaches PC1, the gate loop (G675–F680) connecting PC1 and PC2 is bent upward, and N676 interacts with F617 on the switch loop (Fig. 5a). Note

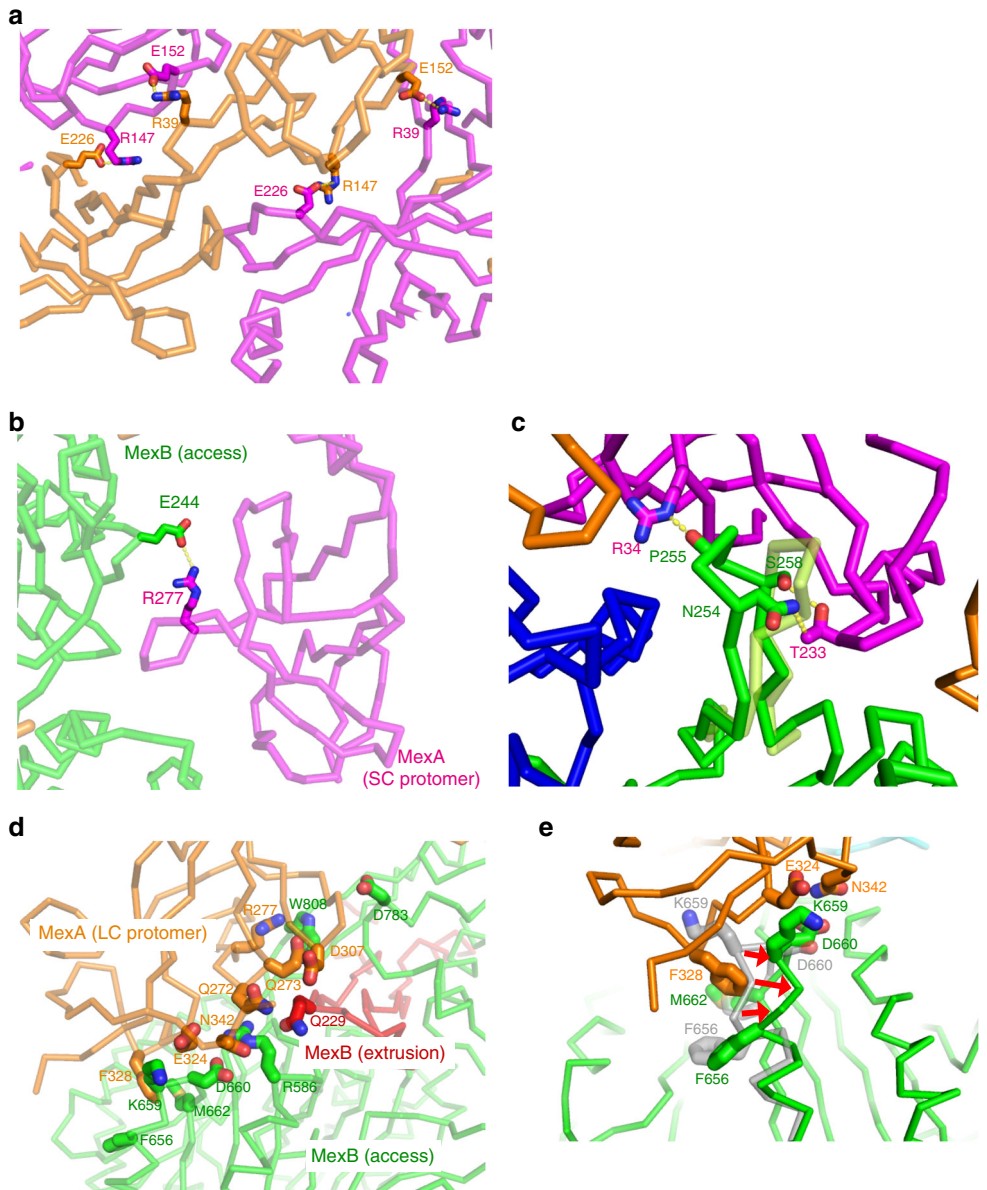

**Fig. 3** Detailed views of the MexA–MexA and MexA–MexB interactions. **a** Detailed view of the lipoyl domains of MexA [shown in magenta (SC protomer) or orange (LC protomer)]. Side chain atoms of residue pairs identified in our model of complex formation are shown as stick models. **b** Close-up view of the interaction between the FL domain in MexB (green) and the SC protomer (magenta). The key loop in the crystal structure (PDB ID: 3w9i) is shown in transparent lemon. Side chain atoms of critical residues identified in the complex formation experiment are shown as stick models. **c** Interaction between the porter domain of MexB (green) and the SC protomer (magenta). **d**, **e** Interaction between the PC1 domain (green) and LC protomer (orange) viewed from the side of PC1 (**d**) or top of PC1 (**e**). The adjacent MexB protomer is shown in red. The shifted helix from the crystal structure of MexB in the Access state is shown in gray. Red arrows show a movement of shifted helix

that a similar interaction is observed in the Extrusion state in the MexB crystal structure. Consequently, in the cryo-EM structure, all three MexB protomers are closed toward the outside, representing a state that cannot accommodate drug molecules. Based on this evidence, we concluded that this is a resting state, in which drug efflux does not occur.

**MexAB–OprM in the presence of drug**. To determine how the presence of drug affects the structure of MexAB–OprM, we added novobiocin, an aminocoumarin antibiotic, to the sample to create a grid for cryo-EM. Data collection and single-particle analysis were performed in the same manner as for the drug-free state, and maps with resolutions of 3.5 Å (0° state) and 3.6 Å (60° state)

were acquired (Supplementary Fig. 4). Overall, each structure is very similar to the corresponding drug-free structure, with large differences only at the gate loop and the PC2 domain of the Bind protomer of MexB. In the presence of novobiocin, the corresponding MexB protomer is similar to the crystal structure; the PC2 opened outward, the cleft between the PC1 and the PC2 opened, and the gate loop was stretched and descended. In addition, a density corresponding to novobiocin appears in the distal binding pocket (Supplementary Fig. 8).

## Discussion

In our cryo-EM structure, OprM exhibits two types of binding states in both the presence and absence of the drug. When OprM

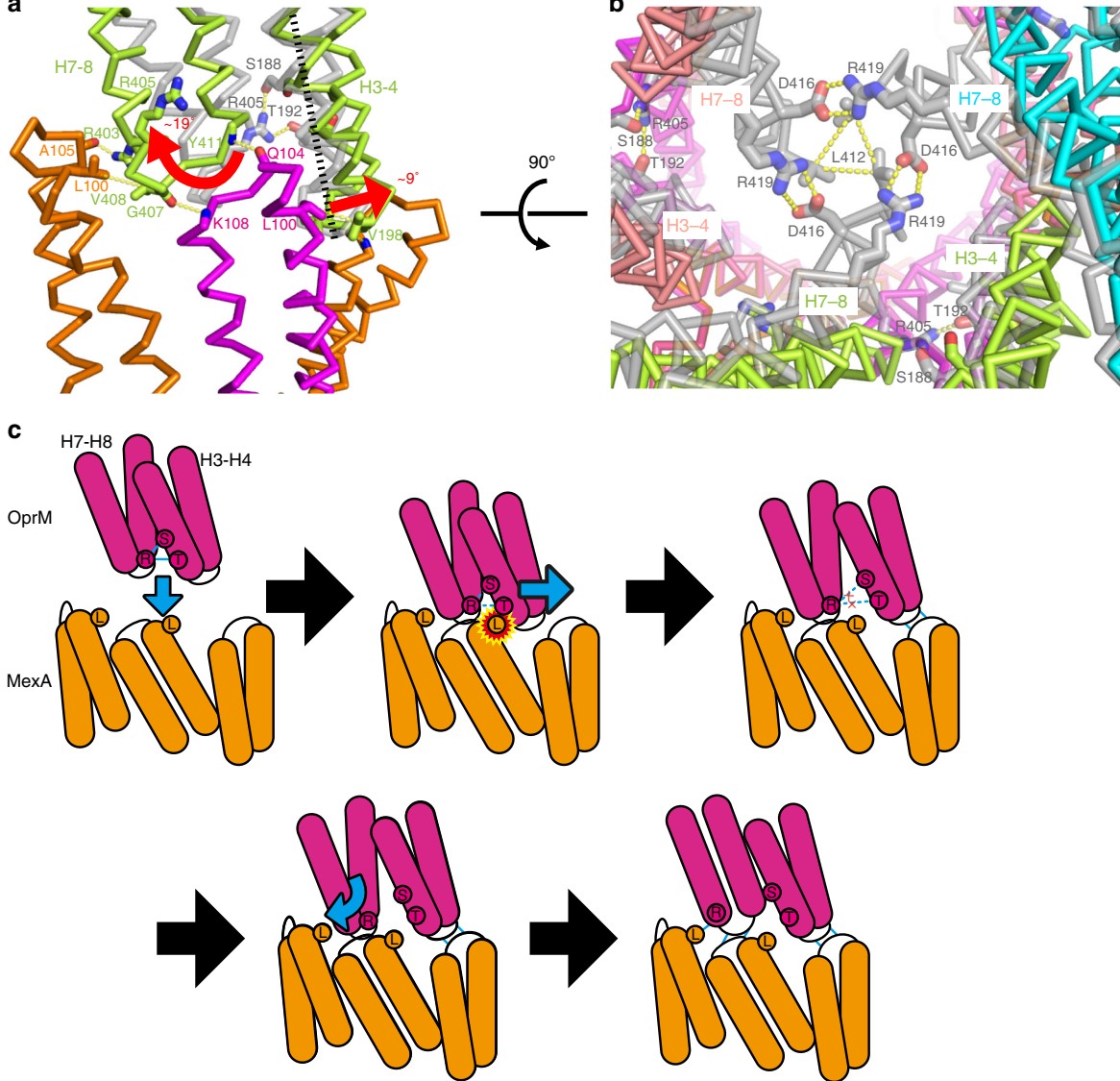

**Fig. 4** A model of OprM channel opening. **a**, **b** Superposition of the OprM crystal structure (PDB ID: 3d5k, gray) and cryo-EM structure (0° state, colored according to Fig. 1) viewed from the periplasmic space (**a**) or from outside the cell (**b**). Red arrows indicate helix movements of H3–H4 or H7–H8 from the closed crystal structure to the open cryo-EM structure. **c** Schematic cartoon of the channel-opening mechanism of OprM. Enclosed letters with circles indicate critical residues for channel opening and complex formation: L indicates L100 in MexA, and S, T, and R indicate S188, T192, and R405 in OprM, respectively

states were separated in single-particle analysis, the ratio of particles in the 0° versus 60° states was 45:55 (in the absence of drug) and 58:42 (in the presence of novobiocin), suggested an overall ratio of nearly 1:1. MexB, which is embedded in the inner membrane, and MexA, which is anchored to the inner membrane, can move freely on the inner membrane, whereas OprM, which penetrates the hard outer membrane and peptidoglycans[28], is remarkably restricted in terms of movement and rotation. The presence of two binding modes in OprM is likely to increase the chances of contact between MexA and OprM.

In the crystal structure of OprM alone, the periplasmic gate is closed, whereas in the complex structure obtained by cryo-EM, the gate is open. Previous study showed that MexAB is required for opening OprM[29]. Based on our native complex structure and mutagenesis experiments, we propose the following mechanism by which OprM opens while binding MexA:

The side face of the H3 helix in OprM and residue L100 in MexA clash when the crystal structure of OprM and the cryo-EM

structure are superposed (Fig. 4a). Because our functional analyses showed that L100 is essential for complex formation, and H3–H4 opens slightly outward in the cryo-EM structure, clash between L100 and H3–H4 may be responsible for the initial movement. After H3–H4 is pushed out by L100, and the tip of H3–H4 interacts with the confronting α-hairpin tip of MexA (Fig. 4c, top centre). Subsequently, hydrogen bonds between H3 and H7 (S188 or T192–R405) in the protomer are broken by opening H3–H4 (Fig. 4c, top right). Meanwhile, H7–H8 becomes unstable, causing the hydrophobic interaction of L412 and the D416–R419 salt bridge between protomers to be broken; consequently, H7–H8 starts to rotate dramatically (Fig. 4c, bottom left). This rotation is stopped by a side interaction with MexA, which is fixed by a tip-to-tip interaction with H3–H4 on the left adjacent protomer (due to a hydrophobic interaction of L100 in MexA or the hydrogen bond between A105 in MexA and R403 in OprM). Finally, by interacting with the side of H3–H4, a tip-to-tip interaction forms between H7–H8 and the fixed MexA α-hairpin

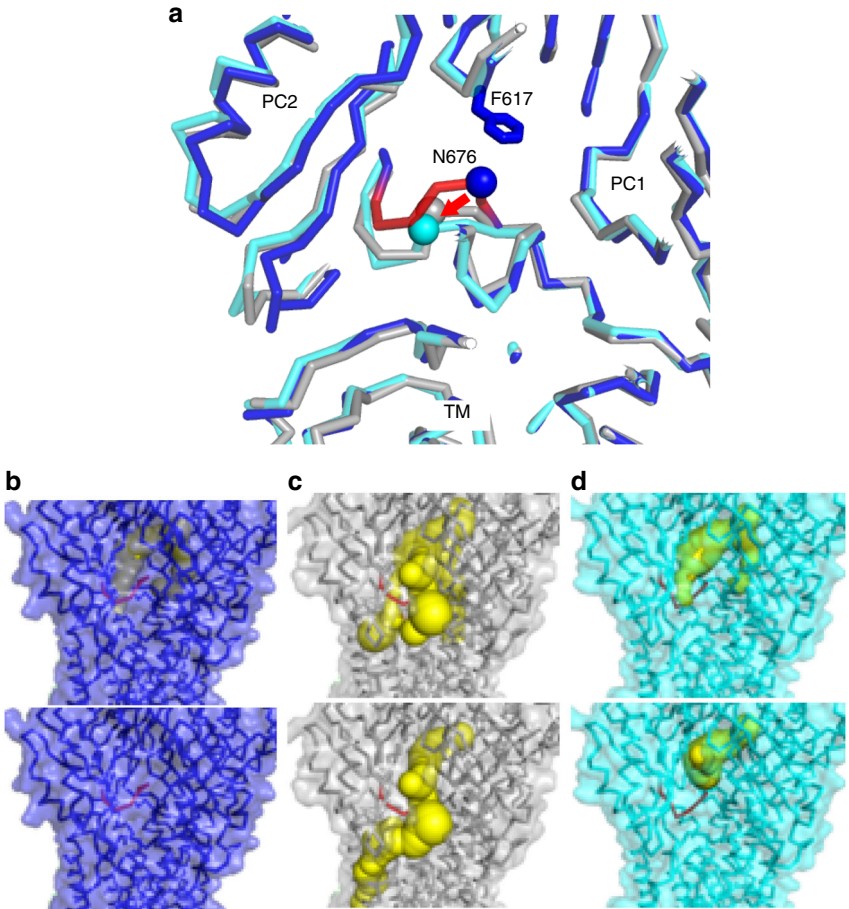

**Fig. 5** Conformational change of MexB from the Resting state to the Binding state. **a** Superposition of the Resting state in the cryo-EM structure (blue), the Binding state in the crystal structure (PDB ID: 3w9i, gray), and the Binding state in the cryo-EM structure (cyan). Gate loops are colored in red. Red arrows indicate conformational changes of the gate loop. **b–d** Visualization of channels to the distal binding pocket, generated using caver v. 3.0.1[45]. Shown are the Resting state in the cryo-EM structure (**b**), the Binding state in the crystal structure (**c**), and the Binding state in the cryo-EM structure (**d**). Channels are shown as yellow spheres with diameters greater than 1.6 Å (top panels) and 2.0 Å (bottom panels). Atomic models are shown in ribbon diagram, and surface views are transparent with the same color as in **a**. Gate loops are colored in red

(Fig. 4c, bottom right). As a result, OprM with a fully opened periplasmic gate is formed. The opened and closed structures of AcrAB–TolC from *E. coli* have been reported[17]. Although we could not observe a closed complex in our single-particle analysis, the structures of both MexAB–OprM and AcrAB–TolC in which OMF are opened are quite similar and previous studies show that MexB can form a chimeric complex with AcrA and TolC[18,30,31], therefore, the mechanism of opening the OMF might be similar in both AcrA–TolC and MexA–OprM.

Based on the results of this study, we propose a mechanism for MexAB–OprM complex formation and a model for drug efflux, as follows. In the complex-formation experiment, SEC peak shifts were observed for MexA and MexB, but not for MexA and OprM or OprM and MexB (Supplementary Fig. 9), suggesting that the MexA–MexB complex is more stable than MexA–OprM complex and MexA might form a complex with MexB prior to OprM. MexA is sequentially bound by MexB via interactions between the MP and β-barrel domains in MexA and the FL domain in MexB, and the bound MexA engages in mutual interactions between lipoyl domains to form a hexameric ring, resulting in formation of the MexA–MexB intermediate (Fig. 6a). When OprM comes in contact with the ring formed by the α-hairpin domain of the MexA hexamer, MexA L100 clashes with the H3 helix in OprM, and the H3–H4 helix is pushed outward (Fig. 6b). After that, as described above, the H7–H8 helix rotates, the interaction between

the OprM trimer and the MexA hexamer is completed, and the periplasmic gate opens (Fig. 6c, d). When the surrounding drug concentration becomes high, the gate loop of MexB shifts downward, and the binding pocket is opened to the molecular surface. After that, MexB ejects drugs into the tunnel of MexA–OprM via a functional rotation mechanism. When the concentration of drugs in this tunnel becomes higher than their concentration outside the cell, they diffuse out of the cell via the concentration gradient (Fig. 6e). As the concentration of drug in the cell decreases, the drug entrance of the binding protomer of MexB is closed by the structural change of the gate loop, and the complex shifts from the Binding state to the Resting state. Under these conditions, because the pump is completely closed to the periplasm, backflow of drugs is prevented. When the concentration of drug in the environment rises, the resting protomer again undergoes a structural change to the Binding state, and drugs are taken in and released. This study provides insights into RND-type multidrug efflux pump; however, these results were based on snapshots of the pump in a particular conformation. Therefore, further dynamic structural study such as MD simulation might be needed for verifying our proposed mechanism.

## Methods
**Expression and purification of MexA, MexB, and OprM**. MexA, MexB, and OprM were expressed and purified as previously described[7,9,11], with slight

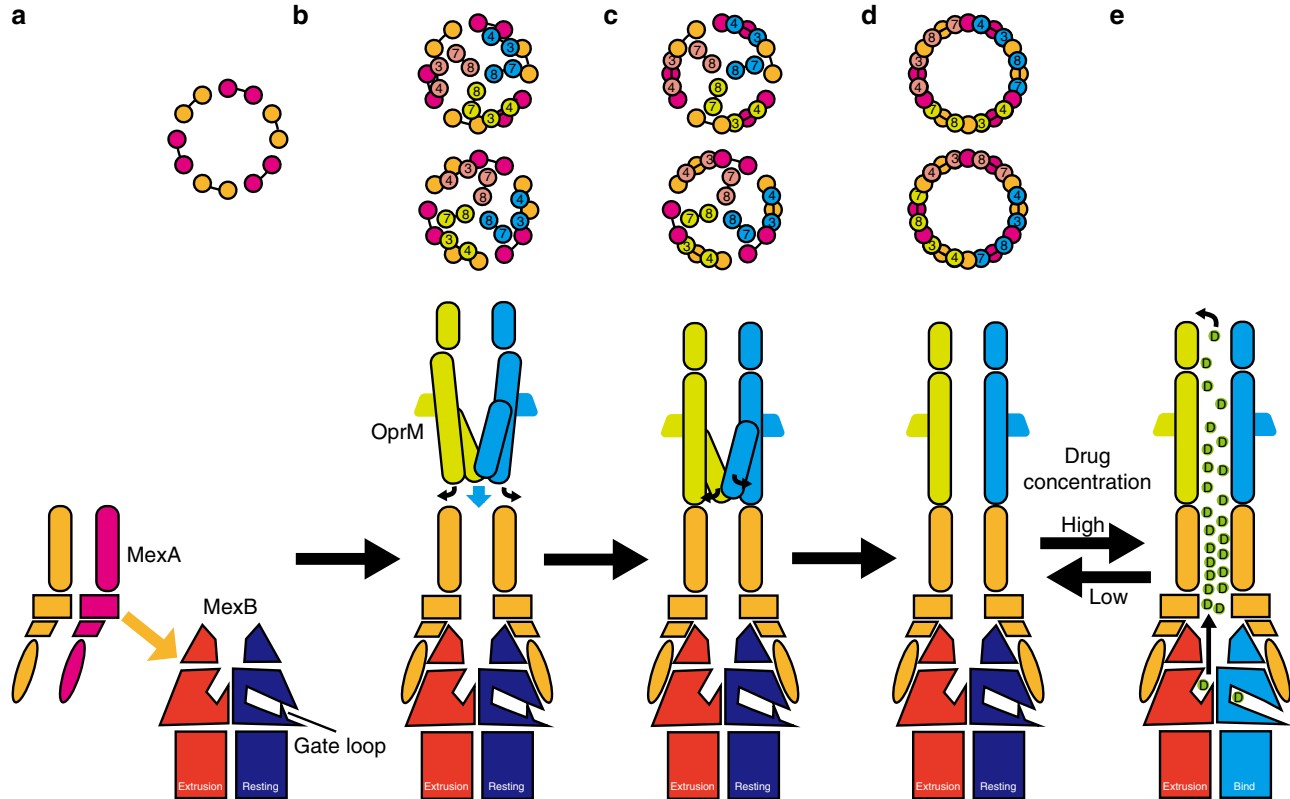

**Fig. 6** Proposed model for complex formation and drug efflux by MexAB–OprM. Sectional views of the MexA–OprM interaction surface section in the 0° state (top panels) and 60° state (middle panels), and views from the periplasmic space of the complex in the 0° state (bottom panels). Each protomer is colored as in Figs. 1 or 5. **a** Six MexA protomers bind to the MexB trimer and form a hexameric tube. **b** The closed OprM trimer interacts with the MexA hexamer by opening the H3–H4 helices. **c** The H7–H8 helices revolve outward and are trapped against the side of MexA via hydrophobic interactions. **d** Formation of the fully opened MexAB–OprM pump is completed. **e** Depending on the drug concentration, the resting protomer in MexB changes its conformation, resulting in formation of the Binding state. Drugs are taken into MexB and pass through the MexA–OprM tube, and are then ejected outside the cell according to their concentration gradients

modifications. All the primers used for cloning are listed in Supplementary Table 2. The gene encoding MexA (a.a., 2–360), which lacks the region containing the signal peptide and the first cysteine, was cloned into pET28 (b+) vector with an N-terminal 6xHis-tag followed by a TEV-protease cleavage site. The resultant plasmid was transformed into BL21-RILP(DE3) (Stratagene, Supplementary Table 3). The bacterial cells were cultured in the LB medium, and the protein expression was induced with 0.4 mM IPTG. Collected cells were suspended in buffer A (50 mM Na-phosphate [pH 7.4], 300 mM NaCl) supplemented with 10 mM imidazole and 1 μM phenylmethylsulfonyl fluoride (PMSF), and then disrupted by sonication and centrifuged at 39,000 g for 30 min. The supernatant was purified with Ni-NTA (QIAGEN), which was eluted with 250 mM imidazole. The 6xHis-tag was removed with TEV-protease, and the de-tagged sample was subjected to a Superdex200 16/60 column (GE healthcare) with buffer B (20 mM Na-phosphate [pH 7.4], 150 mM NaCl). Peak fractions were concentrated to ~50 mg mL$^{-1}$ using a VIVASPIN 20 mL (10,000 MWCO).

The gene encoding full-length MexB was cloned into pET22 (b+) vector with a C-terminal 6xHis-tag, and the resultant plasmid was transformed into C43(DE3) (OverExpress, Supplementary Table 3). The bacterial cells were cultured in the TB medium, and protein expression was induced with 1.2 mM IPTG. Collected cells were disrupted in a French press (SMT CO., LTD.), and debris was removed by centrifugation at 10,000 g for 10 min at 4 °C. Membrane fractions were collected by ultracentrifugation at 235,000 × g for 1 h at 4 °C (45 Ti rotor, Beckman). Collected membrane fractions were washed with high-salt buffer (50 mM Na-phosphate [pH 7.4], 1 M NaCl), and then collected by ultracentrifugation. This wash step was repeated three times. The washed membranes were resuspended in buffer A supplemented with 30% (v/v) glycerol, flash frozen in liquid nitrogen, and stored at −80 °C. Thawed membranes were suspended in buffer A, and then 1% (w/v) n-dodecyl-β-D-maltoside (DDM, Anatrace) was added for additional washing. After ultracentrifugation at 164,000 × g for 1 h at 4 °C (SW32 Ti rotor, Beckman), soluble impurities were discarded. The precipitate was resuspended in buffer A and supplemented with 2% (w/v) DDM and 40 mM imidazole (pH 7.4). The mixture was stirred at 4 °C for 1 h, and the insoluble fraction was removed by ultracentrifugation, as described above. The soluble fraction was subjected to Ni-chelating Sepharose (GE Healthcare) in an Econo-column (Bio-Rad). The

resin was washed with buffer A containing 150 mM imidazole and 0.1% (w/v) 7-cyclohexyl-1-heptyl-β-D-maltoside (CYMAL-7, Anatrace), and then purified MexB was eluted with buffer A containing 350 mM imidazole and 0.02% CYMAL-7. Eluted samples were gathered, concentrated using a SPIN-X 20 mL (100,000 MWCO), and subjected to a Superdex200 16/60 column with buffer C (buffer B supplemented with 0.02% CYMAL-7). Peak fractions were concentrated to ~25 mg mL$^{-1}$.

The gene encoding full-length OprM was cloned into pET21(b+) vector with a C-terminal 6xHis-tag, and the resultant plasmid was transformed into C43(DE3). The bacterial cells were cultured in 2x YT medium, and the protein expression was induced with 1.2 mM IPTG. Disruption, membrane fractionation, and first wash were performed as described above for MexB. The inner membrane fraction was solubilized with 2% (v/v) Triton X-100 at 4 °C for 20 min. Solubilized inner membrane was removed by ultracentrifugation, and then washed once with 50 mM Na-phosphate [pH 7.4] supplemented with 5% glycerol. The washed membranes were resuspended with buffer A supplemented with 30% (v/v) glycerol, flash frozen in liquid nitrogen, and stored at −80 °C. Thawed membranes were suspended with buffer A supplemented with 2.5% n-octyl-β-D-glucoside (OG, Anatrace) and imidazole (pH 7.4) to 20 mM. The mixture was stirred at 4 °C for 1 h, and the insoluble fraction was removed by ultracentrifugation at 235,000 × g for 1 h at 4 °C (50.2 Ti rotor, Beckman). The soluble fraction was subjected to Ni-NTA in an Econo-column. The resin was washed with buffer A containing 40 mM imidazole and 0.1% CYMAL-7, and then purified OprM was eluted with buffer A containing 250 mM imidazole and 0.02% CYMAL-7. Eluted fractions were gathered and subjected to a Superdex200 16/60 column with buffer C. Peak fractions were concentrated to ~10 mg mL$^{-1}$ using a SPIN-X 20 mL (100,000 MWCO).

**Reconstruction of MexAB–OprM.** Purified MexA, MexB, and OprM were mixed at a molar ratio of 3:1:1 in buffer C, and the mixture was dialyzed against buffer D (20 mM Na-citrate, 300 mM KCl, 0.02% CYMAL-7) at 4 °C. We note that we used excess MexA to improve the efficiency of reconstitution. Unreconstructed proteins were removed by SEC on a Superdex200 16/60 column in buffer D. Peak fractions were concentrated with a SPIN-X 20 mL (100,000 MWCO).

**Detergents removal and replacement with Amphipol**. Reconstructed MexAB–OprM (~4 mg) was precipitated by mixing buffer D supplemented with 20% (v/v) PEG-3350 at a volume ratio of 1:2 and then centrifuging at 20,400 $g$ for 30 min at 4 °C. After removal of the supernatant, 250 μL of 50 mM HEPES-K (pH 7.5) containing 20 mg of Amphipol A8–35 (Anatrace) was added to the pellet, and the sample was incubated at 4 °C for 4 h with gentle rotation. Subsequently, ~125 mg of Bio-Beads SM2 (Bio-Rad) was added to the sample, which was rotated at 4 °C overnight. The beads were removed with a poly-prep column (Bio-Rad). The sample was subjected to a Superose6 Increase 10/300 column (GE Healthcare) with 50 mM HEPES-K (pH 7.5). Peak fractions were concentrated with an Amicon Ultra-0.5 mL 100 K (Merck Millipore).

**EM data acquisition**. For the apo-state, 2 μL of sample solution (9.1 mg mL$^{-1}$) was applied to a glow-discharged holey carbon film (Quantifoil 1.2/1.3R, 300-mesh Mo grid). The grid was blotted for 6 s and flash frozen in liquid ethane using a Vitrobot Mark IV (FEI). The datasets were collected on a Titan Krios G2 (FEI) operated at 300 kV, equipped with an FEI Falcon II direct detector. Images were recorded at a nominal magnification of ×75,000 (corresponding to a pixel size of 0.875 Å). Thirty-two frames were recorded for exposure times of 2 s, with defocus value ranging from −1.25 to −3.0 μm; the total dose was 40 e$^-$ Å$^{-2}$.

For the novobiocin-binding state, the sample (10.2 mg mL$^{-1}$) was mixed with 100 mM novobiocin in 500 mM HEPES-K (pH 7.5) at a volume ratio of 9:1 1 h before grid preparation. As a result, the final concentrations of protein and novobiocin were 9.2 mg mL$^{-1}$ and 10 mM, respectively. Grids were prepared as described above for the apo-state. The datasets were collected on a Titan Krios G2 operated at 300 kV, equipped with an FEI Falcon III direct detector (linear mode). Images were recorded at a nominal magnification of ×59,000 (corresponding to the pixel size of 1.125 Å). Thirty-two frames were recorded for exposure times of 2 s, with defocus value ranging from −1.25 to −2.5 μm; the total dose was 40 e$^-$ Å$^{-2}$.

**Image processing and 3D reconstruction**. Movies were motion-corrected using MOTIONCORR2 (version 0130217)[32] with dose fraction. CTF values were estimated with Gctf (version 1.06)[33]. The following processes were performed in RELION-2.0[34,35]. For the apo-state, 8772 micrographs were used for particle picking (Supplementary Fig. 3). A total of ~20,000 manually picked particles were subjected to two-dimensional (2D) class averaging to create a reference for auto-picking. A total of 535,948 particles were extracted from micrographs by auto-picking, and 2D class averaging was performed to remove false particles. After 2D classification, the remaining 420,182 particles were subjected to 3D classification using an initial model created from a previous 3D reconstruction of MexAB–OprM using data from negative-stained EM single-particle analysis. We performed 3D refinement on the 174,534 particles remaining after 3D classification. Although the cryo-EM map of MexAB–OprM at this step had ~4 Å resolution, several ambiguous regions, including the upper region of OprM and the PC2 and TM regions of MexB, were present. To remove ambiguity, we performed sequential local 3D classification of 155,822 particles, which were subjected to additional whole 3D classification to remove bad particles. First, we performed local classification of the OprM–MexA region, enabling separation of the two binding states of OprM (0° state: 57,499 particles; 60° state: 69,224 particle). Second, we attempted to remove the ambiguity in the MexB region by local classification, but this approach failed because of a symmetry mismatch in the α-hairpin domain of MexA (C6 symmetry) and MexB trimer (pseudo-C3 symmetry). Therefore, we used a symmetry expansion method to apply the correct angles to the particles[36]: The dataset was enlarged threefold by adding 0°, 120°, or 240° to the first Euler angle for each particle (172,347 particles in the 0° state; 207,672 particles in the 60° state), and subjected to local 3D classification of the MexB region. Finally, particle sets with good homogeneity and correct angles were selected judging from the clarity of TM and PC2 (37,971 particles in the 0° state; 42,338 particles in the 60° state). These particles were subjected to 3D refinement, yielding maps with near-atomic resolution (for the 0° state, 4.21 Å in unmask and 3.72 Å in mask; for the 60° state, 4.55 Å in unmask and 3.93 Å in mask). To calculate maps with better resolution, we subtracted the density corresponding to Amphipol and performed a final 3D refinement. This approach yielded improved maps: 4.12 Å in unmask and 3.64 Å in mask for the 0° state, and 4.17 Å in unmask and 3.76 Å in mask for the 60° state.

For the novobiocin-binding state, we performed single-particle analysis as described above for the apo-state (Supplementary Fig. 4a). A total of 4681 micrographs were used for particle picking, and 902,901 particles were extracted by auto-picking. After 2D classification, 659,742 particles remained, and 445,966 particles were subjected to the first 3D refinement after 3D classification. The calculated map had same ambiguities as the apo-state, so we performed local classification for OprM–MexA, resulting in separation of the 0° state (230,289 particles) and 60° state (215,677 particles). After symmetry expansion and two rounds of local classification for the MexB region, good particle sets with 31,409 (0° state) or 31,466 (60° state) particles were subjected to 3D refinement. In these refinements, EM maps were calculated for the 0° state at 4.04 Å in unmask and 3.60 Å in mask, and for the 60° state, 4.14 Å in unmask and 3.71 Å in mask. After density subtraction and final refinement, resolution was improved to 3.95 Å in unmask and 3.50 Å in mask (0° state) and 4.09 Å in unmask and 3.60 Å in mask (60° state).

**Model building and map sharpening and structural validation**. Models of fully opened OprM trimer and MexA hexamer were prepared using molecular dynamics flexible fitting (MDFF)[37]. Closed OprM (PDB ID: 3d5k) was fitted into the calculated map (apo-state, 0° state) by rigid-body fitting using Chimera[38], followed by flexible fitting using MDFF. Six MexA protomers (PDB ID: 2v4d; https://doi.org/10.2210/pdb2V4D/pdb, B chain) were fitted into the map using Chimera, and subsequently fitted using MDFF. The crystal structure of MexB (PDB ID: 3w9i; https://doi.org/10.2210/pdb3W9I/pdb) was simply fitted into the map using Chimera. The coordinate files were gathered and subjected to one round of real-space refinement by phenix.real_space_refine[39] and reciprocal space refinement by phenix.refine[40]. The atomic model that performed this process was used for local B-factor sharpening of the cryo-EM map using the program locscale[41]. Final models were obtained after several rounds of manual correction by coot[42] and real-space refinement against the locally sharpened maps (Supplementary Table 1). Local resolution validation was performed with ResMap (Supplementary Figs. 3, 4)[43]. The pathways for drug efflux in MexB were estimated using caver 3.0.1.

**Complex-formation experiment**. Plasmids for each mutant were prepared by a standard PCR method. All the primers used for mutation are listed in Supplementary Table 2. Expression, purification, and reconstruction of each mutant was performed as described above (Supplementary Fig. 5). Reconstruction experiments for MexA–MexB, MexA–OprM and MexB–OprM were performed as described for the reconstruction of MexA–OprM, except for the omission of OprM, MexB, and MexA, respectively (Supplementary Fig. 9). The flow-rate of SEC was fixed at 0.5 mL min$^{-1}$.

**The in vivo drug-resistance assay**. The drug-resistance assay was performed using a methodology previously described[44]. The genes of *mexA*, *mexB*, and *oprM* were subcloned into the pMMB67HE vector. The mutants were prepared with a standard PCR method. All the primers used for mutation are listed in Supplementary Table 2. The resultant plasmids were transformed into W3104Δ*acrABD*. Bacterial cells were cultured overnight in 3 mL of the LB medium supplemented with 100 μg mL$^{-1}$ of ampicillin at 37 °C. All of the cultures were adjusted to an OD$_{590nm}$ of 1.0 with the LB medium and then serially diluted with tenfold dilutions (10$^{-1}$–10$^{-6}$). For each dilution series, 4 μL of each strain was plated onto LB plates supplemented with 100 μg mL$^{-1}$ of ampicillin and an additional antibiotic: none, 5 μg mL$^{-1}$ of chloramphenicol, 25 μg mL$^{-1}$ of novobiocin, 5 μg mL$^{-1}$ of erythromycin, 0.5 μg mL$^{-1}$ of minocycline, or 10 μg mL$^{-1}$ of rhodamine 6 G. All plates were incubated at 37 °C overnight and then images were recorded.

**Reporting summary**. Further information on experimental design is available in the Nature Research Reporting Summary linked to this article.

## Data availability

Data supporting the findings of this manuscript are available from the corresponding author upon reasonable request. A reporting summary for this Article is available as a Supplementary Information file. Cryo-EM density maps of the apo-state MexAB–OprM have been deposited in the Electron Microscopy Data Bank under accession codes EMD-9695 (0° state), and EMD-9696 (60° state). Atomic coordinates have been deposited in the Protein Data Bank under accession codes 6IOK (0° state), and 6IOL (60° state).

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

## Acknowledgements

We thank S. Murakami for providing the *E. coli* strain W3104Δ*acrABD*, T. Nakae for providing the pMMB67HE vector and plasmids encoding *mexA*, *mexB*, and *oprM*, and S. Nagata for constructing the expression vectors. This work was supported by JSPS KAKENHI Grant number JP25291014 and MEXT KAKENHI Grant number JP22121006 (to E.Y.) and the Basis for Supporting Innovative Drug Discovery and Life Science Research (BINDS) from the AMED.

## Author contributions

K.T., R.Y., A.N. and E.Y. designed the project. K.T., R.Y. and E.I.-I. performed the protein preparation. N.M., K.I. and S.M. collected the EM data. K.T. performed the structural analysis. R.Y. and E.I.-I. performed functional analysis. K.T., R.Y., E.Y. and A.N. wrote the paper.
