## [Peer Review File · Nature Communications]

Reviewers' Comments:

Reviewer #1:

Remarks to the Author:

This work presents the structure of the fully assembled tripartite multidrug efflux pump MexAB-OprM from *P. aeruginosa*, and achieves near atomic resolution by single particle analysis cryo-EM. The manuscript describes the interaction of OprM-MexA, MexA-MexB, and the whole complex, and the results are validated with site mutations including the complex formation and drug efflux in vivo assays. A model is proposed for OMF-MFP-RND complex formation and drug efflux, in which MexA forms a complex with MexB first and opens the gate of OprM, followed by modulation of MexB conformation according to drug concentration. The report presents both Apo- and drug-bound states of the full complex, leading to the observation of conformational change in MexB but not in OprM/MexA. The report shows that the reconstitution of OprM-MexAB is straight forward. The strategy used for cryo-EM data processing to resolve the problem of ambiguity in two states of OprM, and three states of MexB, including using local classification and symmetry expansion, and final subtract of amphipol, are essential for reaching this high resolution. This is an excellent study and there are a few comments that might hopefully be helpful for the authors:

The conclusion of "first formation of MexA-MexB" is based on SEC. Line 281-283: MexAB complex is detected by SEC rather than other binary complex (Supplemental fig 9) is not definitive evidence to conclude "MexA-MexB complex is formed first". In consequence, figure 6 (a) may need to be modified. What is the in the first elution peak in the orange curve?

The opening mechanism of OprM/OMF has been an important topic, in the structure here the OprM gate is already open in the complex even in the apo-form and MexB in "resting" state? While drug added, only conformational change in MexB. (line 236-246) In figure 4 their mechanism depicts a pushing or steric effect, but there could be conformational selection from dynamic fluctuations. Whilst Fig. 4 is certainly helpful in illustrating the authors' point about the OprM channel opening, a movie containing a morph between the two states would help the reader to visualise the mechanism.

Line 122: What is the predicted map resolution in the MexA/OprM interface? Unless it is around or below 3Å, statements about the details of hydrogen bonding should be perhaps made with greater caution. It could be a good idea to show the map of the area with the fitted model in a supplementary figure. The authors could make a note of this issue in text and explicitly refer to their mutational analysis in the following paragraphs as intended to validate the proposed bonding.

Minor points:

1. The PC1 domain of MexB is discussed many times in the paper, but it would be helpful to introduce this domain, e.g. at line 59-64.
2. Line 73 – the AcrAB-TolC structure also include disulphide linked AcrA-AcrB without an engineered liker.
3. Line 115-118 and line : need to note "see figure 1 and 2"
4. Line 118-120: text not clear: RMSD comparison of the periplasmic tip of OprM between the two states? Tip of which protomer? Are three protomers the same in the map? The same problem for "a-hairpin domain" of MexA. If this is the case, it might be confusing to write "in both states",

because it sounds like the comparison is being made within each state.

5. Line 130: Distribution of the two states within the cells concluded only by the particle numbers needs the caveat that the freezing conditions may conceivably have biased the distribution. There is a distinct possibility that in a living bacterial cell this proposition would not hold true, as the authors have not tested for the distribution of the states and rely solely on in vitro data. A rephrasing of this very strong proposition would be advisable.

6. Figure 1 label and figure legends need to be improved. Label the bottom row, please.

7. Figure 3e Labelling is crowded, please improve the view and labels. Red arrow shows the movement?

8. Figure 4. a and b: presentation of the grey crystal structure makes it difficult to distinguish the cryo-EM structure, please improve (e.g. transparency/cartoon bond thickness/text color etc).

9. Figure 5. Errors in legend line 466-467: both (b) and (c) are from cryo-EM structure? Presumably one of them is crystal structure.

10. Figure 6 e: please change the color for "bind" state to distinguish from "resting" state.

11. Methods line 566: Please confirm MexA: MexB: OprM is mixed in a ratio 3:1:1, although the stoichiometry is "2:1:1" (line 96).

12. Poor resolution of supplemental figures 3 and 4, labels in (e) (i) not readable.

13. Supplemental figure 4, labels in the figures not consistent with the legends.

14. Line 25: The sentence as written seems to suggest that presence of MexAB-OprM pump results in a nosocomial infection. It might be better to rephrase.

15. Line 33: Not sure what the authors mean by "complex formation drug efflux"; should it perhaps read "drug efflux by complex formation", or "complex-formation drug efflux"?

16. Line 54: For the general reader, it might not be clear which of the three MexAB-OprM components are the MFP, the OFP and the RND mentioned in the previous paragraph.

17. Line 111: Do the authors mean "denote" instead of "donate"?

18. Line 193 "Note that these interactions hardly differed..." This is a little unclear and might be rewritten.

19. Line 196 "flops" might be better as "shifts"

20. Line 255: Is this statement supported by authors' observations or previously published work?

21. Line 258: As the open conformation of OprM certainly exists in the crystalline form, the question remains whether it could exist in solution. Did the authors at any point try to find or analysed particles of OprM only, with the pump detached, in their cryo-EM datasets?

22. Line 283: This still does not exclude a possibility for the pre-formation of the complex with OprM and MexA, the SEC would only strictly speaking suggest that of the three possibilities OprM + MexA, OprM + MexB and OprM + MexAB the third is the most stable in the conditions of the

experiment.

23. Line 423: For clarity, it could be helpful to label the states in the figure as well.

24. Line 466: Authors define both panels (b) and (c) as depicting the resting state in the Cryo-EM structure, is this intentional? Additionally, to help guide the reader, the authors could label the states in the figure as well.

25. Line 621: Could the authors provide more information on their symmetry expansion strategy? More specifically, whether care was taken not to allow the averaging of the non-symmetrical parts of their pseudo-symmetric object. Was at any point a subtraction of the symmetrised particles from the non-symmetrised considered? This seems specifically pertinent with respect to the possible multiple conformational states of MexB.

26. Line 636 typo "as the as"

Reviewer #2:

Remarks to the Author:

This is a very strong structural study of the efflux pump MexAB-OprM from *P.aeruginosa*. The pump is the major contributor to the antibiotic resistance and the structural info is important. There are several other cryo-EM structures of the homologous AcrAB-TolC pump from *E.coli*. The authors found different interactions between MexA and OprM and propose a model for the assembly of the pump.

The work is convincing and the paper will influence thinking in the field.

There are a few weaknesses that are important to address.

1. The figure legends for Supplementary information lack critical details and should be revised. For example, Fig S2 and Fig S5 legends should include conditions of SEC such as buffer composition, flow rate etc.

2. Fig. S6 is expected to show MICs but shows growth inhibition experiments at a single drug concentration. It is important that the author measure MICs and report them in a Table. MIC values are critical for reproducibility and comparison with previous studies. The description of these experiments in the main text and in the Method is incorrect as no MIC measurements were done. If the authors decide to keep the Fig. S6, the legend should contain important details to understand the experiment, such conditions and concentrations of used antibiotics. Abbreviations should not be used with the names of antibiotics unless they are standard in the field.

3. In Figs, S3, S4 and S6, the plots are impossible to read because of the low quality images.

4. The text is overloaded with abbreviations and difficult to read. It should be revised for clarity.

Reviewer #3:

Remarks to the Author:

This manuscript report on the reconstitution and structural analysis of the assembled tripartite efflux pump MexAB-OprM from *Pseudomonas aeruginosa*. This is an important advance as it is the first time a tripartite efflux pump other than AcrABZ-TolC have been solved to near-atomic resolution. This is a comprehensive and well written manuscript that would be of great interest to people in the community and the wider field.

Even though this is not a high resolution crystal structure, the overall structure of the assembled complex is undoubtedly an accurate representation of the assembly of MexAB-OprM. Similar to data for the AcrABZ-TolC complex there is also no interaction between the OMF and the IMP in this structure. Overall there is a very good correlation between functional data and the structure. For instance, when residues that are predicted to be involved in the interaction between the individual

proteins in the complex were mutated the result was a disruption of pump assembly. The structure of the overall complex also align with the predicted stoichiometry of 2 : 1 : 1 for MexA : MexB : OprM . It also reflected in the assembled structure with MexA forming a ring consisting of six MexA proteins in two different conformations similar to the structure of AcrA in the assembled AcrABZ-TolC complex.

The structures of all the individual proteins (MexA, MexB and OprM) in the complex have already been solved by X-ray crystallography. Some differences are reported between these crystal structures and the structures of the concomitant proteins in the assembled structure. For example, the crystal structure of OprM by itself revealed that the periplasmic aperture is closed, while the structure of OprM in this assembled complex is open and ready for drug export. This is also different from the situation in the AcrAB-TolC assembly, where it is proposed that the TolC initially joins the complex in the closed state and then opens via rearrangement in the AcrA driven by conformational changes in AcrB.

The differences in the crystal structure of OprM and that of OprM in the complex is not difficult to explain as it makes sense for an OMF to be closed when not in a complex with its cognitive partners; an open OMF by itself would be problematic for the bacterium. Additionally, just like TolC, OprM also forms a complex with other transporters e.g. MexXY hence would not always be in a complex.

The more intriguing issue is the proposed differences in assembly between MexAB-OprM and AcrAB-TolC suggested by the assembled structure reported in this study. This could possibly a result of the fact that engineered AcrAB proteins were used to solve the structure of AcrAB-TolC as suggested by the authors. It could also be a result of intrinsic differences between TolC and OprM as the two proteins have only a low sequence homology (Figure S1C). However, I am not sure that the latter is the case as the overall assembly process were reported to be the same for both complexes. It is also already known that the individual proteins of the two complexes are promiscuous and that MexB can form a functional complex with AcrA and TolC (J Bacteriol., 2008, 190:691-8 and Biochem J., 2010, 430:355-64). Also, in a separate study the reconstitution of native MexAB-OprM, AcrAB-TolC and interspecies AcrA-MexB-TolC complexes revealed a common assembly mechanism between AcrAB-TolC and MexAB-OprM. The authors should comment on this issue. This work does not currently refer to the studies on the functional reconstitution of the MexAB-OprM complex (Nat Commun., 2015, 22;6:6890 and Nat Commun., 2016, 7:10731). The authors should add a discussion and comparison of the data from this study to that of the functional data of the assembled pump.

The overall export mechanism proposed based on the structures of the apo-complex and a novobiocin-bound complex suggests some novel features and differences compared to previously published mechanisms. However, these structures are snapshots of the transporter in a particular conformation, hence the proposed transport mechanism could be verified in later follow-up studies.

Minor things:

Line 33: "...complex formation **and** drug efflux."

Lane 476 in the legend to Fig 6: as not has

Reviewers' comments:

Reviewer #1 (Remarks to the Author):

*This work presents the structure of the fully assembled tripartite multidrug efflux pump MexAB-OprM from *P. aeruginosa*, and achieves near atomic resolution by single particle analysis cryo-EM. The manuscript describes the interaction of OprM-MexA, MexA-MexB, and the whole complex, and the results are validated with site mutations including the complex formation and drug efflux in vivo assays. A model is proposed for OMF-MFP-RND complex formation and drug efflux, in which MexA forms a complex with MexB first and opens the gate of OprM, followed by modulation of MexB conformation according to drug concentration. The report presents both Apo- and drug-bound states of the full complex, leading to the observation of conformational change in MexB but not in OprM/MexA. The report shows that the reconstitution of OprM-MexAB is straight forward. The strategy used for cryo-EM data processing to resolve the problem of ambiguity in two states of OprM, and three states of MexB, including using local classification and symmetry expansion, and final subtract of amphipol, are essential for reaching this high resolution. This is an excellent study and there are a few comments that might hopefully be helpful for the authors:*

First of all, we would like to thank the referee for his/her valuable and helpful comments.

The conclusion of “first formation of MexA-MexB” is based on SEC. Line 281-283: MexAB complex is detected by SEC rather than other binary complex (Supplemental fig 9) is not definitive evidence to conclude “MexA-MexB complex is formed first”. In consequence, figure 6 (a) may need to be modified. What is the in the first elution peak in the orange curve?

We agree with this comment and the 22nd comment in the minor points. Therefore, we corrected the text at line 299-302 (Line 281-283 of previous manuscript) as follows:

SEC peak shifts were observed for MexA and MexB, but not for MexA and OprM or OprM and MexB (Supplementary Fig. 9), suggesting that the

MexA–MexB complex is more stable than MexA-OprM complex and MexA might form a complex with MexB prior to OprM.

The first elution peak in the orange curve in Supplementary figure 9 indicates OprM. We labeled each peak in Supplementary figure 9.

The opening mechanism of OprM/OMF has been an important topic, in the structure here the OprM gate is already open in the complex even in the apo-form and MexB in “resting” state? While drug added, only conformational change in MexB. (line 236-246) In figure 4 the mechanism depicts a pushing or steric effect, but there could be conformational selection from dynamic fluctuations. Whilst Fig. 4 is certainly helpful in illustrating the authors’ point about the OprM channel opening, a movie containing a morph between the two states would help the reader to visualise the mechanism.

Yes, the OprM gate was already open in the complex, in the presence or absence of drug. We added the movies “Supplementary Movie 1 and 2”, as the reviewer suggested.

Line 122: What is the predicted map resolution in the MexA/OprM interface? Unless it is around or below 3Å, statements about the details of hydrogen bonding should be perhaps made with greater caution. It could be a good idea to show the map of the area with the fitted model in a supplementary figure. The authors could make a note of this issue in text and explicitly refer to their mutational analysis in the following paragraphs as intended to validate the proposed bonding.

Is the predicted map the local resolution map? If so, the resolution around the MexA/OprM interface is around 3Å. We added the Supplementary Figure 11 that shows the potential maps of the region of the MexA/OprM interface, it shows the side chain features in that region. We added the text at line 119 as follows:

” Local resolution validation shows that the contact face region between OprM and MexA has ~3 Å resolution which allows us to fit good atomic models into the maps (Supplementary Fig.3 e and h). ”

Minor points:

1. *The PC1 domain of MexB is discussed many times in the paper, but it would be helpful to introduce this domain, e.g. at line 59-64.*

We added the description of 4 subdomains of porter domain of MexB at line 59-61 as follows:

“A porter domain further consists of four subdomains; PN1, PN2, PC1 and PC2, which translate and rotate per subdomain, thereby altering drug accessibility.”

2. *Line 73 – the AcrAB-TolC structure also include disulphide linked AcrA-AcrB without an engineered liker.*

We added a phrase to include disulfide linked AcrA-AcrB in the sentence “However, because these studies used genetically engineered MFP-RND” as follows:

“However, because these studies used genetically engineered or disulfide linked MFP-RND”

3. *Line 115-118 and line : need to note “see figure 1 and 2”*

We added a sentence “(Figs. 1 and 2)” to the end of line 124 (Line 118).

4. *Line 118-120: text not clear: RMSD comparison of the periplasmic tip of OprM between the two states? Tip of which protomer? Are three protomers the same in the map? The same problem for “a-hairpin domain” of MexA. If this is the case, it might be confusing to write “in both states”, because it sounds like the comparison is being made within each state.*

We removed the phrase “in both states” and re-wrote the sentence “In both states, the C α RMSD between periplasmic tips of the OprM protomer (S188–E214: H3-H4; Y396–F422: H7-H8) is 1.1 Å, and the C α RMSD between α -hairpin domains of hexameric MexA (A74–F134) is 0.33 Å. The contact surface areas between the MexA hexamer and OprM trimer are 2992 Å² (0° state) and 2983 Å² (60° state).” as follows :

“The C α RMSD between periplasmic tips of the OprM protomer **in two states** (S188–E214: H3-H4; Y396–F422: H7-H8) is 1.1 Å, and the C α RMSD between α -hairpin domains of hexameric MexA **in two states** (A74–F134) is 0.33 Å. The contact surface areas between the MexA hexamer and OprM trimer are 2992 Å² (0° state) and 2983 Å² (60° state).”

5. Line 130: Distribution of the two states within the cells concluded only by the particle numbers needs the caveat that the freezing conditions may conceivably have biased the distribution. There is a distinct possibility that in a living bacterial cell this proposition would not hold true, as the authors have not tested for the distribution of the states and rely solely on in vitro data. A re-phrasing of this very strong proposition would be advisable.

We agree with this comment. We re-phrased the sentence “we conclude that these two binding....” to “it might be considered that these two binding....”.

6. Figure 1 label and figure legends need to be improved. Label the bottom row, please.

Labels in figure 1 were moved to the bottom and corrected, and figure legends were corrected.

7. Figure 3e Labelling is crowded, please improve the view and labels. Red arrow shows the movement?

Labels in Figure 3e were made the similar colors as each chain, and changed to bold.

Yes, the red arrow shows the movement. Therefore, we added a sentence in the figure legends as follows:

“The red arrow shows a movement of shifted helix”

8. Figure 4. a and b: presentation of the grey crystal structure makes it difficult to distinguish the cryo-EM structure, please improve (e.g. transparency/cartoon bond thickness/text color etc).

Labels in Figure 4 a and b were made the similar colors as each chain, and changed to bold.

9. Figure 5. Errors in legend line 466-467: both (b) and (c) are from cryo-EM structure? Presumably one of them is crystal structure.

Thank you for pointed out our mistake. (b) is the crystal structure. We corrected the description of (b).

10. Figure 6 e: please change the color for “bind” state to distinguish from “resting” state.

We changed the color for “binding” state in the figure 6e according to the reviewer’s comment.

11. Methods line 566: Please confirm MexA: MexB: OprM is mixed in a ratio 3:1:1, although the stoichiometry is “2:1:1” (line 96).

The sentence “Purified MexA, MexB, and OprM were mixed at a molar ratio of 3:1:1” is correct. When the purification of MexAB-OprM complex, MexA: MexB: OprM was mixed in a ratio 3:1:1. Nevertheless, the stoichiometry of MexA: MexB: OprM in the structure of MexAB-OprM complex was 2:1:1.

12. Poor resolution of supplemental figures 3 and 4, labels in (e) (i) not readable.

Supplemental figures 3 and 4 were replaced with high resolution figures.

13. Supplemental figure 4, labels in the figures not consistent with the legends.

Thanks. We corrected the legends.

14. Line 25: The sentence as written seems to suggest that presence of MexAB-OprM pump results in a nosocomial infection. It might be better to rephrase.

We changed the sentence “In Pseudomonas aeruginosa, MexAB–OprM plays a central role in multidrug resistance by ejecting various drug compounds, resulting in serious nosocomial infections.” as follows:

“In *Pseudomonas aeruginosa*, MexAB–OprM plays a central role in multidrug resistance by ejecting various drug compounds, **which is one of causes of** serious nosocomial infections.”

15. Line 33: Not sure what the authors mean by “complex formation drug efflux”; should it perhaps read “drug efflux by complex formation”, or “complex-formation drug efflux”?

We changed the phrase “complex formation drug efflux” as follows: “drug efflux and complex formation”

16. Line 54: For the general reader, it might not be clear which of the three MexAB-OprM components are the MFP, the OFP and the RND mentioned in the previous paragraph.

We added MFP, RND and OFP after MexA, MexB and OprM, respectively, in the sentence of line 50-51, as follows: “The crystal structures of the three proteins constituting MexAB–OprM, MexA (**MFP**), MexB (**RND**), and OprM (**OMF**), have already been solved⁷⁻¹².”

17. Line 111: Do the authors mean “denote” instead of “donate”?

Thanks. We corrected it.

18. Line 193 “Note that these interactions hardly differed...” This is a little unclear and might be rewritten.

We changed the sentence as follows: “Note that **there is little difference in these interactions....**”.

19. Line 196 “flops” might be better as “shifts”

We changed to “shifts”.

20. Line 255: Is this statement supported by authors’ observations or previously published work?

We added references showing of mobility of membrane proteins on inner or outer membranes.

21. Line 258: *As the open conformation of OprM certainly exists in the crystalline form, the question remains whether it could exist in solution. Did the authors at any point try to find or analysed particles of OprM only, with the pump detached, in their cryo-EM datasets?*

No, we did not try to analysis particles of OprM only in solution. But, the crystal structure of OprM in previously published work showed the close conformation.

22. Line 283: *This still does not exclude a possibility for the pre-formation of the complex with OprM and MexA, the SEC would only strictly speaking suggest that of the three possibilities OprM + MexA, OprM + MexB and OprM + MexAB the third is the most stable in the conditions of the experiment.*

We agree with this comment. Therefore, we corrected the text at line 299-302 (Line 281-283) as follows:

SEC peak shifts were observed for MexA and MexB, but not for MexA and OprM or OprM and MexB (Supplementary Fig. 9), suggesting that the MexA–MexB complex is more stable than MexA–OprM complex and MexA might form a complex with MexB prior to OprM.

23. Line 423: *For clarity, it could be helpful to label the states in the figure as well.*

We wrote the label of the states in the Fig. 1, and corrected the panel label.

24. Line 466: *Authors define both panels (b) and (c) as depicting the resting state in the Cryo-EM structure, is this intentional? Additionally, to help guide the reader, the authors could label the states in the figure as well.*

We mistook the figure legends of the panel (c). The panel (c) depicts the binding state in the crystal structure. We corrected the figure legends of panel (c). Thanks for your suggestion. We tried to label the states in the panels (b), (c) and (d) of the figure, but we thought the state names, example for “the binding state in the Cryo-EM structure”, were too long. Therefore, we did not label them in the figure.

25. Line 621: *Could the authors provide more information on their symmetry expansion strategy? More specifically, whether care was taken not to allow the averaging of the non-symmetrical parts of their pseudo-symmetric object. Was at any point a subtraction of the symmetrised particles from the non-symmetrised considered? This seems specifically pertinent with respect to the possible multiple conformational states of MexB.*

In the first map calculated without non-symmetry and symmetry, the map of the PC2 and TM regions was more ambiguous than that of the other regions in the MexB. We thought that the map was an average of some states of MexB and that it was necessary to decide the orientation of MexB using the PC2 and TM domains of the non-symmetry parts.

To show the detailed region for judging whether the orientation of MexB is the right, we changed the phrase the “the bottom region” to “PC2 and TM regions” at the line 434 and added the phrase “judging from the clarity of TM and PC2” at the line 446.

26. Line 636 typo “as the as”

Thanks. We corrected it.

Reviewer #2 (Remarks to the Author):

This is a very strong structural study of the efflux pump MexAB-OprM from P.aeruginosa. The pump is the major contributor to the antibiotic resistance and the structural info is important. There are several other cryo-EM structures of the homologous AcrAB-TolC pump from E.coli. The authors found different interactions between MexA and OprM and propose a model for the assembly of the pump.

The work is convincing and the paper will influence thinking in the field.

There are a few weaknesses that are important to address.

1. *The figure legends for Supplementary information lack critical details and should be revised. For example, Fig S2 and Fig S5 legends should include conditions of SEC such as buffer composition, flow rate etc.*

We added the buffer composition and flow rate in Fig S2. legends. The conditions of SEC in Fig S5 were same with Fig S2 (b), therefore we added a sentence in Fig S5. legends as follows: “The conditions of SEC are as described in Fig. S2.”

2. *Fig. S6 is expected to show MICs but shows growth inhibition experiments at a single drug concentration. It is important that the author measure MICs and report them in a Table. MIC values are critical for reproducibility and comparison with previous studies. The description of these experiments in the main text and in the Method is incorrect as no MIC measurements were done. If the authors decide to keep the Fig. S6, the legend should contain important details to understand the experiment, such conditions and concentrations of used antibiotics. Abbreviations should not be used with the names of antibiotics unless they are standard in the field.*

Thanks for pointing out. Our experiments were drug resistance assays that were performed by Adler etc. (Biochemistry 43, 518-525, 2004) but were not MIC experiments as you say. Therefore, we corrected the word “MIC experiment” to “drug resistance assay” in text and Fig. S6. We added a sentence and the reference “Biochemistry 43, 518-525, 2004” at the line 491 as follows:

“The drug resistance assay was performed as described previously⁴⁴.”

We added the conditions and concentrations of used antibiotics and rewrote the abbreviations of antibiotics to their common names in Fig. S6.

3. *In Figs, S3, S4 and S6, the plots are impossible to read because of the low quality images.*

Figs S3, S4 and S9 were improved to high resolution images.

4. *The text is overloaded with abbreviations and difficult to read. It should be revised for clarity.*

We rewrote the abbreviations of antibiotics to their common names in the text and the figures.

Reviewer #3 (Remarks to the Author):

This manuscript report on the reconstitution and structural analysis of the assembled tripartite efflux pump MexAB-OprM from Pseudomonas aeruginosa. This is an important advance as it is the first time a tripartite efflux pump other than AcrABZ-TolC have been solved to near-atomic resolution. This is a comprehensive and well written manuscript that would be of great interest to people in the community and the wider field.

Even though this is not a high resolution crystal structure, the overall structure of the assembled complex is undoubtedly an accurate representation of the assembly of MexAB-OprM. Similar to data for the AcrABZ-TolC complex there is also no interaction between the OMF and the IMP in this structure. Overall there is a very good correlation between functional data and the structure. For instance, when residues that are predicted to be involved in the interaction between the individual proteins in the complex were mutated the result was a disruption of pump assembly.

The structure of the overall complex also align with the predicted stoichiometry of 2 : 1 : 1 for MexA : MexB : OprM . It also reflected in the assembled structure with MexA forming a ring consisting of six MexA proteins in two different conformations similar to the structure of AcrA in the assembled AcrABZ-TolC complex.

The structures of all the individual proteins (MexA, MexB and OprM) in the complex have already been solved by X-ray crystallography. Some differences are reported between these crystal structures and the structures of the concomitant proteins in the assembled structure. For example, the crystal structure of OprM by itself revealed that the periplasmic aperture is closed, while the structure of OprM in this assembled complex is open and ready for drug export. This is also different from the situation in the AcrAB-TolC assembly,

where it is proposed that the TolC initially joins the complex in the closed state and then opens via rearrangement in the AcrA driven by conformational changes in AcrB.

The differences in the crystal structure of OprM and that of OprM in the complex is not difficult to explain as it makes sense for an OMF to be closed when not in a complex with its cognitive partners; an open OMF by itself would be problematic for the bacterium. Additionally, just like TolC, OprM also forms a complex with other transporters e.g. MexXY hence would not always be in a complex.

The more intriguing issue is the proposed differences in assembly between MexAB-OprM and AcrAB-TolC suggested by the assembled structure reported in this study. This could possibly a result of the fact that engineered AcrAB proteins were used to solve the structure of AcrAB-TolC as suggested by the authors. It could also be a result of intrinsic differences between TolC and OprM as the two proteins have only a low sequence homology (Figure S1C). However, I am not sure that the latter is the case as the overall assembly process were reported to be the same for both complexes. It is also already known that the individual proteins of the two complexes are promiscuous and that MexB can form a functional complex with AcrA and TolC (J Bacteriol., 2008, 190:691-8 and Biochem J., 2010, 430:355-64). Also, in a separate study the reconstitution of native MexAB-OprM, AcrAB-TolC and interspecies AcrA-MexB-TolC complexes revealed a common assembly mechanism between AcrAB-TolC and MexAB-OprM. The authors should comment on this issue. This work does not currently refer to the studies on the functional reconstitution of the MexAB-OprM complex (Nat Commun., 2015, 22;6:6890 and Nat Commun., 2016, 7:10731). The authors should add a discussion and comparison of the data from this study to that of the functional data of the assembled pump.

In our study, a closed OprM in the MexAB-OprM complex like a closed TolC in the complex with disulphide linked AcrA-AcrB could not be observed. Since each protomer of MexB in the MexAB-OprM complex in the absence of drug formed asymmetric conformation as in the crystal structure, OprM in the complex may have been open. We agree with a common assembly mechanism between AcrAB-TolC and MexAB-OprM.

We think that OprM is opened by a rigid tube formation of six MexA protomers binding to MexB and forms a more stable MexAB-OprM complex. Therefore, we added the text to describe more precisely as follows at line 291-296: “The opened and closed structures of AcrAB–TolC from *E. coli* have been reported¹⁷. Although we could not observe a closed complex in our single-particle analysis, the structures of both MexAB–OprM and AcrAB–TolC in which OMF are opened are quite similar and previous studies shows that MexB can form a chimeric complex with AcrA and TolC^{18,29,30}, therefore, the mechanism of opening the OMF might be similar in both AcrA–TolC and MexA–OprM.”

Thank you for pointed out that we lost the study about the studies on transport activity of the MexAB-OprM complex *in vitro* (Nat Commun., 2015, 22;6:6890). Therefore, We added the sentence at line 271-272 as follows: “Previous study showed that MexAB is required for opening OprM²⁹. “

The overall export mechanism proposed based on the structures of the apo-complex and a novobiocin-bound complex suggests some novel features and differences compared to previously published mechanisms. However, these structures are snapshots of the transporter in a particular conformation, hence the proposed transport mechanism could be verified in later follow-up studies.

We agree that these structures are snapshots of the transporter in a particular conformation. Therefore, we added the sentence “This study provided new insight into RND-type multidrug efflux pump, however, these results were based on snapshots of the pump in a particular conformation. Therefore, further dynamic structural study such as MD simulation might be needed for verifying our proposed mechanism.” in the end of the text as the reviewer’ comment.

Minor things:

*Line 33: “...complex formation **and** drug efflux.”*

We changed to “drug efflux and complex formation”

Lane 476 in the legend to Fig 6: as not has

Thanks. We corrected it.

Reviewers' Comments:

Reviewer #1:

Remarks to the Author:

The revised manuscript has been improved. There are a few minor points that the authors might consider:

Line 60 Might read better as

"A porter domain further consists of four subdomains; PN1, PN2, PC1 and PC2, which translate and rotate during the transport process, thereby altering drug accessibility."

line 67 might read better as

"...suggesting that the mechanisms of action of RND family members may have diverged."

Statement line 76 not accurate and could be deleted with little loss of information

"However, because these studies used genetically engineered or disulfide linked MFP-RND fusion proteins for structural analysis, it was difficult to discuss about detailed interactions between each protein. "

Change line 79 from "Moreover, although 3D..." to read "Although 3D..."

Line 146 should read "determine the in vivo drug resistance."

Typo line 178 Y411

Line 193 Might wish to note here that MacA also forms hexamers in the pump assembly ; see Fitzpatrick et al (2017) Nature Microbiology

Line 221 "Comparation between"-> "Comparison of"

Line 267 references 27 and 28 suitable to support point?

Reviewer #3:

Remarks to the Author:

The revised paper is now ready for publication.

First of all, we would like to thank the referees for their assistance reviewing our manuscript and for providing helpful suggestions.

REVIEWERS' COMMENTS:

Reviewer #1 (Remarks to the Author):

The revised manuscript has been improved. There are a few minor points that the authors might consider:

Line 60 Might read better as

"A porter domain further consists of four subdomains; PN1, PN2, PC1 and PC2, which translate and rotate during the transport process, thereby altering drug accessibility."

We changed the sentence "A porter domain further consists of four subdomains; PN1, PN2, PC1 and PC2, which translate and rotate per subdomain, thereby altering drug accessibility." according to the reviewer's comment as follows:

"A porter domain further consists of four subdomains; PN1, PN2, PC1 and PC2, which translate and rotate during the transport process, thereby altering drug accessibility."

line 67 might read better as

"...suggesting that the mechanisms of action of RND family members may have diverged."

We changed the sentence "... suggesting that the mechanisms of action of RND family members are not uniform." according to the reviewer's comment as follows:

"... suggesting that the mechanisms of action of RND family members may have diverged."

Statement line 76 not accurate and could be deleted with little loss of information

"However, because these studies used genetically engineered or disulfide linked MFP-RND fusion proteins for structural analysis, it was difficult to discuss about detailed interactions between each protein. "

We agree that the sentence "it was difficult to discuss about detailed interactions between each protein." is not accurate, because the genetically engineered or disulfide linked AcrAB-ToIC structure shows the interaction between MFP (AcrA) and OMF (ToIC). Therefore, we changed the sentence "However, because these studies used genetically engineered or disulfide linked MFP-RND fusion proteins for structural analysis, it was difficult to discuss about detailed interactions between each protein." as follows:

"These studies used genetically engineered or disulfide linked MFP-RND fusion proteins for structural analysis."

Change line 79 from "Moreover, although 3D..." to read "Although 3D..."

We changed "Moreover, although 3D..." to "Although 3D..." according to the reviewer's comment.

Line 146 should read "determine the in vivo drug resistance."

Thank you for your correction. We changed the sentence "determine the in vivo drug resistance assay" to "determine the in vivo drug resistance."

Typo line 178 Y411

Thanks. We corrected it.

Line 193 Might wish to note here that MacA also forms hexamers in the pump assembly ; see Fitzpatrick et al (2017) Nature Microbiology

Thank you for your suggestion. We added the phrase "and in the tripartite complex" at the last sentence of "Interaction between each MexA protomer" subsection and the reference "*Fitzpatrick et al (2017) Nature Microbiology* ".

Line 221 "Comparison between"-> "Comparison of"

Thanks. We changed "Comparation between" to "Comparison of"

Line 267 references 27 and 28 suitable to support point?

We changed the two references, "Spector, J. et al. Biophysj 99, 3880, (2010)" and "Weiß, K. et al. Biophysj 105, 455, (2013)" to a reference "Yamamoto, K. et al. Sci. Rep. 1–10 10.1038/srep21909 (2016)". This reference showed that dynamics of AcrB were different in present or absent of TolC. AcrB formed a complex with TolC was stationary. That's because TolC might be restricted by outer membrane and peptidoglycan layer. Therefore, We modified the sentence "OprM, which penetrates the hard outer membrane, is remarkably restricted in terms of movement and rotation" as follows:

"OprM, which penetrates the hard outer membrane and peptidoglycans²⁸, is remarkably restricted in terms of movement and rotation"

Reviewer #3 (Remarks to the Author):

The revised paper is now ready for publication.